# Short-Term Effects of the COVID-19 Outbreak on Consumer Perceptions of Local Food Consumption and the Local Agri-Food Sector in Austria

**Laura Maria Wallnoefer** *[ID] **and Petra Riefler** [ID]

Department of Economics and Social Sciences, Institute of Marketing and Innovation,
University of Natural Resources and Life Sciences Vienna, 1180 Vienna, Austria
* Correspondence: laura.wallnoefer@boku.ac.at

**Abstract:** Disruptions in agri-food systems caused by crises, such as the COVID-19-pandemic, reveal the vulnerability of global food supply chains. Such crises might consequently impact consumer perceptions about the relevance of local food production and consumption. In this light, this study aims to (i) identify whether the COVID-19 outbreak led to short-term changes in perceptions about local food consumption and (ii) capture how the role of local agri-food systems is perceived in times of crisis. For the first purpose, this study analyzes two waves of survey data collected from an Austrian sample (*n* = 351) to compare pre-and post-COVID-19 levels of consumer values, beliefs, and attitudes towards local food. For the second purpose, the paper assesses consumer perceptions about the reliability and resilience of the local agri-food sector in response to the COVID-19 outbreak. The results reveal that while consumer perceptions driving local food consumption at an early phase of the pandemic remained stable at large, the perceived relevance of the local agri-food sector attenuated. Consumers showed strong beliefs in the local agriculture as a reliable and trustworthy partner during the pandemic guaranteeing food supply security. Based on these findings, the paper discusses how these insights into consumer perceptions in response to macro-level disruptions might help to better understand short-term demand-side implications of other forms of external crises affecting local food production and supply. Finally, the paper provides recommendations for practitioners and avenues for future research to determine implications from a long-term perspective.

**Keywords:** local food; consumers; disruptions; COVID-19; perceptions; agri-food-systems; agriculture; crises; downstream effects



## 1. Introduction

Crises such as the COVID-19 pandemic and, most recently, the war in Ukraine have exposed the vulnerability of global food supply chains [1,2]. With the outbreak of the pandemic, several categories of the food supply chain, including fresh fruit and vegetables, bakery items, and food grains, became seriously compromised [3]. The protective measures enforced by governments worldwide further led to a decreased access to abundant, nutritiously sufficient, and diverse sources of food [4]. These measures also triggered a global economic shock at a fast pace, conveying high recessions for many countries [5]. The disruption following the outbreak of the COVID-19 pandemic consequentially led to raising concerns regarding the resilience of supply chains to shocks [6–8], food security [9,10], and food safety [10–12]. As a result of food safety concerns, consumers increased their consumption of domestic products [12]. In parallel, from a supply perspective, shorter supply chains became essential due to their high supply chain responsiveness [13,14], which is a key dimension of resilience, i.e., the ability of a system to continually change and adapt in response to stressors and shocks [15]. Overall, the interest in local and regional food has substantially increased [16] and is likely to further grow in the short to medium term post-COVID-19 [6].

Local food systems are often considered more resilient to shocks, due to their ability to draw on nearby social support and local resources and the more direct interaction between food producers and consumers [17–20]. Many voices state that the pandemic highlights the relevance of local production and short food supply chains being less affected by international restrictions [21]. It also brought attention to the importance of the agri-food sector as a pillar of the economy [22]. Governments in many countries thus called on consumers to consume more local food aiming at supporting the local suppliers and the economy overall (as, for example, done by the Austrian Federal Ministry of Agriculture Regions and Tourism (BMLRT), 2020). While local food consumption was already a well-established trend prior to the COVID-19 outbreak [6,23,24], the pandemic might have further attenuated relevant consumer perceptions driving local food consumption. A change in perceptions can bring about a change in the interactions of consumers with other actors within the local agri-food sector, i.e., producers and retailers, which can have relevant implications for supply chain resilience [6,14,15]. Favorable consumer perceptions of local agriculture were, for example, recently linked to a rising interest in short food supply chains and alternative food networks [25].

So far, research has touched upon the changing consumption patterns in response to the COVID-19 crisis, although it has primarily focused on changes regarding purchase frequency and type of purchased foods [26,27], choice criteria and new concerns related to food safety [10,28], as well as food preparation [29]. Thereby, most studies measure the respective changes by assessing self-reported perceptions of changes in one's consumption patterns in response to the crises. The observed change thus lacks a baseline, which limits insights into the degree of change in objective terms [30]. Researchers, often due to the limited availability of pre- and current COVID-19 data [31], refer to the disruptions following the pandemic outbreak as the general cause for the reported changes. Such implicit and generalized assumptions about the cause of change pose some limitations when aiming to identify more detailed mechanisms inducing the respective changes in consumption patterns [31]. The understanding of these mechanisms can be improved by capturing subjective perceptions of consumers about changing conditions [15] within specific sectors (i.e., the local agri-food sector), which can have direct implications for them. In light of the political emphasis on local food supply paired with the recognized relevance of the domestic agri-food sector, consumer perceptions regarding health and environmental aspects of local food and its role in the local economy might have also changed [6]. Jointly capturing consumer perceptions about changing conditions in the local agri-food sector and consumer perceptions that have been established drivers of local food consumption before the pandemic might further add to the understanding of underlying mechanisms underlying the changes in consumption patterns. Insights into pandemic-related changes in consumer values, beliefs, and attitudes relevant to local food demand, however, are currently scant and often lack a baseline assessment prior to the pandemic outbreak.

In this light, we aim to (i) increase understanding of COVID-19-induced changes in consumers' perceptions about local food consumption and (ii) assess consumers' perceptions of the role of the local agri-food sector in times of crises. We use Austria as an exemplary country that imports high quantities of fruits and vegetables despite its high self-sufficiency rate [32]. For the first purpose, we use survey data collected from an Austrian consumer sample in two waves, i.e., before the COVID outbreak (November 2019) and after the first wave of lockdowns (May 2020). In particular, we compare pre-and post-outbreak levels of selected consumer values, beliefs, and attitudes which were identified as relevant drivers of local food demand by previous research. For the second purpose, we assess the following consumer perceptions about the local agri-food sector in terms of: (a) supply security and system relevance, (b) status during the pandemic, and (c) trust and reliability for future crises. In this vein, we attempt to capture potentially relevant internal and external drivers to changes in local food consumption by assessing both perceptions that have been established drivers of local food consumption and perceptions of the local agri-food sector in times of crises. While the assessment of consumer perceptions at an early stage of

the pandemic in May 2020 requires replications to validate observed changes over time, the early-stage insights on how consumers perceived initial disruptions might be valuable for the design of immediate measures to withstand shocks. In sum, we aim to contribute to the agribusiness literature by increasing understanding of COVID-19 implications for local food supply from a consumer perspective, which might also be indicative of emerging crises other than the pandemic.

In the following, we first provide an overview of the theoretical background and the development of our hypothesis. We then elaborate on the materials used and the methods applied to examine changes in consumer perception about local food consumption and the role of the local agri-food sector during the crisis. We then present and discuss empirical results using relevant literature in the field. Finally, we conclude the paper by providing recommendations for practitioners and directions for research on local food consumption.

## 2. Theoretical Background and Hypothesis Development

Both production and consumption patterns can change due to crises that affect the stability and predictability of contextual factors, such as social or governmental rules and regulations [33,34]. With COVID-19 as a systemic shock, the supply chains connecting producers and consumers became a source of systemic vulnerability [7]. The pandemic consequentially revealed how a shock in one country could have a so-called ripple effect across the world and how a single event could impact multiple geographies and points along the supply chain simultaneously [35]. As agri-food systems are characterized by interdependent groups of actors and feedbacks [35], these ripple effects triggered by major sources of externality, i.e., lockdowns and mobility restrictions imposed by governments, move throughout the system both downward from the producers and upward from the consumers [13]. As such, downstream effects move from producers that experience disruptions, for example, of input supply chains or workers' availability to retailers, vendors, and, eventually, consumers in the form of reduced stability of food availability [13,36,37]. Upstream effects occur as a result of, for example, increased costs of food, disrupted access or loss of income, or panic buying, and can affect vendors, retailers, and, eventually, producers in the form of reduced demand for certain food items [12,13,38]. The crisis caused by the outbreak of the COVID-19 pandemic thereby altered the patterns of food production and consumption [22], which resulted in disruptions within agri-food supply chains both on the supply and demand sides [6,36]. A holistic analysis of the implications of the COVID-19 pandemic for local food consumption on an individual level thus needs to consider consumers as recipients of downstream effects and as senders of upstream effects.

Capturing the dynamic of consumers as recipients and senders can contribute to the understanding of a local agri-food sector's resilience. Resilience as a concept is helpful in the context of the research on COVID-19 implications, as it helps to better understand a fundamentally unpredictable world [15]. For an agricultural system to be resilient it requires a balance between the ability to be efficient in the current context and the ability to re-organize and adapt in response to changes that are unforeseen (and unforeseeable) [15]. By demonstrating how the supply chain responsiveness, as a key dimension of resilience [13, 14], is reflected in consumer perceptions, it is possible to make assumptions about the long-run consumer confidence in these supply chains [6,14]. Such confidence is essential for robust and reliable relationships, which are central to enhancing resilience [6].

Consumers, as receivers of downstream effects, i.e., in the form of reduced stability of food availability, can get an increased sense of uncertainty regarding the reliability and security of the food system [6,13]. In this context, a study showed consumer perceptions of a limited supply of daily necessities negatively impact anxiety levels [39] and mechanisms of trust [40]. The respective uncertainties emerging from the reduced stability in food availability can lead to modified reactions and behavior regarding different aspects of consumers' food consumption [38,41,42]. It is, thus, crucial to capture how consumers as recipients of downstream effects perceive the reliability and supply security of producers and retailers. A relevant aspect in this light includes the perceived resilience of the local

agri-food sector, i.e., regarding food security [14]. Potential changes in the status of the sector that is gaining more attention as an important pillar of the economy [22] can further be assessed. To get insights on potentially rising concerns amidst the crises [34,38] it is further valuable to capture how much consumers trust in the local agri-food sector to manage future crises.

Consumers as senders of upstream effects, i.e., in the form of changing demands for specific food items, can be driven by different values, beliefs, and attitudes that influence their preferences for different food items [43]. The key values, beliefs, and attitudes driving local food consumption [44], derived from international marketing research, range from health-related over environmental to socio-economic drivers (for a review of drivers, see [24]). Extant research accordingly identifies *consumer ethnocentrism* [45], which describes a belief that one should preferably buy domestic products over foreign-made products, and the *healthiness bias*, which describes a belief that domestic products are healthier and of superior taste and quality than imported food [46,47], as two key drivers for local food consumption. In the face of the political emphasis on consuming locally [48], the desire to support local businesses can become more meaningful as a driver for local food consumption [6]. In times where global health is threatened, beliefs and attitudes regarding the healthiness and quality of local food, known as the domestic country bias or healthiness bias [46,49], might also change. Correspondingly, a comparison of general food choice motives before and after the pandemic outbreak showed an increase in the importance of health [50]. Restrictions and economic shocks might be reflected in a change of *price consciousness*, i.e., the degree to which the consumer exclusively focuses on paying a low price [51]. As such, a recent study showed that consumers were concerned about price increases after the outbreak of the pandemic [10]. If they are highly conscious of the price of a product, consumers are less likely to opt for local food [52,53]. The consciousness about the value of environmental protection through one's consumption choices, i.e., *green consumer values* [54], in contrast to the consciousness about the price, is rarely reflected in actual food choices [52,55]. Accordingly, while green consumer values may be high among consumers, these consumers do not accordinglychoose more environmentally friendly food products [44], which indicates a value action gap [56]. Amidst the COVID-19 pandemic, studies reported mixed findings about the role of ethical concerns as food choice motives. Accordingly, one study found that ethical concerns remained predominately stable as a motive [50], while another showed their increasing relevance [57]. Finally, extant research found that consumers' *authenticity perceptions* of domestic agriculture and the food it produces motivate local food choices [44]. Authenticity perceptions are related to aspects such as the continuity, credibility, and symbolism [58] of the local agri-food sector. So far, there is no empirical evidence about possible changes in this perception after the outbreak of the pandemic. The same holds for one's sense of belonging to a local community, traditions, and culture, i.e., the *local identity* [59], which relates to local food choices [44]. As consumers had to adapt their behavior in response to the outbreak of the pandemic, i.e., by consuming more local food [60,61], their perceptions, including those about authenticity and their local identity, might have also been adapted. This notion is based upon the self-perception theory [62], according to which one's past behavior can influence one's perceptions.

In light of the above, we assume a COVID-19-induced change in consumer perceptions about local food consumption, as crises can have severe impacts on individuals' lifestyles, worldviews, and well-being, as well as on their routines [34]. This leads us to the following hypotheses (H1)–(H6):

**Hypothesis 1 (H1).** *The level of consumer ethnocentrism assessed after the outbreak of the COVID-19 pandemic differs from the level assessed before the outbreak of the COVID-19 pandemic.*

**Hypothesis 2 (H2).** *The level of the healthiness bias assessed after the outbreak of the COVID-19 pandemic differs from the level assessed before the outbreak of the COVID-19 pandemic.*

**Hypothesis 3 (H3).** *The level of local identity assessed after the outbreak of the COVID-19 pandemic differs from the level assessed before the outbreak of the COVID-19 pandemic.*

**Hypothesis 4 (H4).** *The level of authenticity assessed after the outbreak of the COVID-19 pandemic differs from the level assessed before the outbreak of the COVID-19 pandemic.*

**Hypothesis 5 (H5).** *The level of green consumer values assessed after the outbreak of the COVID-19 pandemic differs from the level assessed before the outbreak of the COVID-19 pandemic.*

**Hypothesis 6 (H6).** *The level of price consciousness assessed after the outbreak of the COVID-19 pandemic differs from the level assessed before the outbreak of the COVID-19 pandemic.*

To capture how consumers as recipients of downstream effects perceived the local agri-food sector amidst the COVID-19 crisis, we chose a descriptive approach and thus formulated no hypotheses.

## 3. Materials and Methods

### 3.1. Data Collection and Sample Description

In the first wave, we conducted an online survey using the service of a professional panel provider in November 2019 to collect a sample of 499 Austrian consumers (The data on consumer perceptions about local food for the first wave was used in a different form in a previous publication [44]). The representativeness of the sample for the Austrian population was ensured by setting quotas for age (range 18–65), gender, education, and residence accordingly. Following the outbreak of the COVID-19 pandemic, for the second wave, we re-invited the same sample of 499 participants to participate in a follow-up study in May 2020. A total of 351 persons completed this second online survey which built the basis of the analysis. As reported in Table 1, the sample consisted of 51% female participants and 49% male participants who were 51.45 years old on average. Most participants had completed an apprenticeship (40.2%), whereas 14.4% held a degree from a higher education institution. Regarding the residential area, the participants were almost evenly distributed among cities (34.8%), suburbs (35.9%), and rural areas (29.1%).

**Table 1.** Sample Profile.

| Characteristics | | Sample | | Austria * |
|---|---|---|---|---|
| | | Frequency | Percentage (%) | Percentage (%) |
| Gender | Female | 179 | 51.0 | 50.8 |
| | Male | 172 | 49.0 | 49.2 |
| Age (in years) | 15–29 | 41 | 11.7 | 22.4 |
| | 30–44 | 89 | 25.4 | 26.2 |
| | 45–59 | 126 | 35.9 | 29.3 |
| | 60–75 | 95 | 27.1 | 22.1 |
| Highest degree of education | Compulsory school | 20 | 5.7 | 23.7 |
| | Apprenticeship | 141 | 40.2 | 30.7 |
| | Vocational School | 80 | 22.8 | 13.5 |
| | Secondary school | 62 | 17.7 | 16.5 |
| | University degree | 48 | 13.7 | 15.7 |
| Degree of urbanization | Cities | 122 | 34.8 | 32.2 |
| | Suburbs | 126 | 35.9 | 27.7 |
| | Rural areas | 102 | 29.1 | 40.1 |

* Source: [59,60].

### 3.2. Measures of Consumer Perceptions about Local Food Consumption

Both waves measured the six selected consumer variables using established scales. *Consumer ethnocentrism* [45], as a primary driver in the literature, was measured with a short version of the CETSCALE [45]. The *healthiness bias* [49], also referred to as domestic country bias, was assessed with a combination of six items based on Gineikiene et al. [46] and Aprile et al. [63]. Individuals' *local identity* was captured with four items, as applied by

Makri et al. [64] and developed by Tu et al. [59]. The perceived authenticity of domestic agriculture was measured with an adapted version of the brand authenticity scale by Morhart et al. [58]. *Green consumer values*, describing consumers' tendency to express the value of environmental protection through their purchases and consumption behaviors, were measured by five items of the GREEN-scale [54]. To capture a potential barrier to local food consumption, the survey included two items by Koschate-Fischer et al. [65] assessing consumers' *price consciousness* [51].

In wave 2, we additionally added a series of questions to capture how disruptions in the food supply chain following the outbreak of the COVID-19 pandemic were perceived by consumers. Accordingly, the scales assessed consumer perceptions regarding the following: (a) reliability, supply security, and system relevance during the crisis, (b) status during the pandemic, and (c) trust in the local agri-food sector. Finally, we measured the financial vulnerability of the respondents. Regarding (a), we asked participants whether they thought that farmers played an important (system-relevant) role during the crisis, whether Austrian agriculture was a reliable partner in the crisis, and whether it could ensure a stable supply. Regarding (b), we provided the participants with five statements about the continued existence of farms throughout Austria, the assurance of a sufficient supply of basic foodstuffs, the reduction of the existing dependence on foreign food imports, mandatory labeling of the origin of ingredients in processed foods, and the overall change in status of the Austrian agri-food sector. We then asked the respondents to indicate whether the relevance of the stated aspects had changed. Regarding (c), we assessed the respondents' perceptions of the local agri-food sector's ability to ensure a stable supply in the event of future crises.

All variables were measured on 7-point Likert scales and yielded acceptable construct reliability with Cronbach's alpha values above the recommended threshold of 0.7 [66]. All items are listed in Tables A1 and A2 in Appendix A.

*3.3. Data Analysis*

The data analysis consisted of two parts. First, we tested our hypotheses (H1)–(H6) by analyzing the changes in consumer perceptions about local food consumption by comparing the scale means pre- and post-COVID-19 (n = 351) with paired sample t-tests in SPSS 26. Second, we analyzed the descriptive statistics of consumer perceptions about the role of the local agri-food sector in the COVID-19 crisis and examined these perceptions for sociodemographic differences due to gender, age, education, residence, and degree of financial vulnerability. For this purpose, we used independent t-tests to examine the influence of gender and financial vulnerability on focal variables and a series of one-factorial ANOVAs in SPSS 26 to assess the effect of age, education, and residence on focal variables, respectively.

## 4. Results

*4.1. Pre–Post COVID-19 Comparisons of Consumer Perceptions about Local Food Consumption*

Table 2 reports the scale means of each driver (for measurement items, please refer to Table A1 in Appendix A) before/after the pandemic outbreak as well as mean comparisons for the two points in time (n = 351). In absolute terms, the results reveal the *perceived authenticity* of local food to show the highest mean value both before and after the outbreak (mean$_{nov19}$= 5.467, mean$_{may20}$= 5.348), whereas *price consciousness* had the lowest mean value (mean$_{nov19}$= 4.148, mean$_{may20}$= 3.990). A mean comparison over time indicates a significant decrease in both the *perception of authenticity* and *price consciousness* (see Table 2), which supports Hypotheses H4 and H6. Aside from the changes in single items within the different constructs, the analysis revealed no significant changes in the other consumer perceptions, which led to a rejection of H1, H2, H3, and H5 (see Table 2, last column). The respondents' *consumer ethnocentrism*, *healthiness bias*, *local identity*, and level of *green consumer values* thus remained stable throughout the first wave of the pandemic.

**Table 2.** Comparison of consumer perceptions about local food consumption before and after the outbreak of COVID-19.

| Construct | Mean | Std. Deviation | t-Value | *p*-Value |
|---|---|---|---|---|
| Consumer ethnocentrism 19 November | 5.152 | 1.222 | 0.937 | 0.350 |
| Consumer ethnocentrism 20 May | 5.093 | 1.270 | | |
| Healthiness bias 19 November | 5.222 | 1.030 | −0.953 | 0.341 |
| Healthiness bias 20 May | 5.273 | 1.007 | | |
| Local identity 19 November | 5.082 | 1.077 | 1.739 | 0.083 |
| Local identity 20 May | 5.001 | 1.132 | | |
| Authenticity 19 November | 5.467 | 1.028 | 2.666 | 0.008 *** |
| Authenticity 20 May | 5.348 | 1.009 | | |
| Green consumer value 19 November | 4.722 | 1.057 | −0.745 | 0.457 |
| Green consumer value 20 May | 4.752 | 0.953 | | |
| Price consciousness 19 November | 4.148 | 1.470 | 2.754 | 0.006 *** |
| Price consciousness 20 May | 3.990 | 1.393 | | |

Note: All answers were given on 7-point scales ranging from strongly disagree (1) to (7) strongly agree. *** significant at a 0.1% level. The constructs above were also presented in [44].

### 4.2. Consumer Perceptions about the Role of the Local Agri-Food Sector in Times of Crises

The mean values for consumer perceptions of the supply, status, and trustworthiness of the Austrian agri-food sector are shown in Table 3 (for the measurement items, please refer to Table A2 in Appendix A). These results demonstrate that the perception of the supply security and system relevance was overall positive, with 91% of respondents agreeing on the system-relevant role of the domestic agri-food sector during the pandemic. The respondents further perceived the Austrian agri-food sector to have gained status as a result of the pandemic (83% agreed). Consumers' trust in the agri-food sector was also high, as 86% of the respondents relied on the sector's ability to ensure a stable food supply in the event of future crises.

**Table 3.** Descriptive results of consumer perceptions of the local agri-food sector in times of crises.

| Construct | Mean | Std. Deviation |
|---|---|---|
| Supply security and system relevance [a] | 5.972 | 1.015 |
| Status during the pandemic [b] | 5.839 | 0.980 |
| Trust in the local agri-food sector [c] | 5.536 | 1.113 |

Note: All answers were given on 7-point scales. The response scales ranged from 1 [[a] I fully disagree and [c] no trust at all] to 7 [[a] I fully agree and [c] very high trust] as well as from [b] (—) became less important to (+++) became more important.

We used an independent t-test and a one-factorial ANOVA to identify differences in the perceptions regarding sociodemographic characteristics, such as gender, age, education, and residence, as well as regarding financial vulnerability, respectively. The results of the t-test showed that female participants were significantly more likely to state that Austrian agriculture had gained status after the outbreak of the pandemic (mean$_{female}$ = 5.958, mean$_{male}$ = 5.715, *p* = 0.020). The results of the ANOVA showed that age had a significant influence on how consumers perceived (a) supply security and system relevance, and (b) status during the pandemic. Accordingly, the perceived relevance and status increased with the age of the consumers. The rural/urban residence of respondents did not influence perceptions, and the same applied to the degree of financial vulnerability.

### 5. Discussion, Limitations, Conclusions

This study aimed to identify whether the disruptions following the outbreak of the COVID-19 pandemic led to changes in, first, consumer perceptions driving local food

consumption and, second, the perceived relevance of the domestic agri-food sector in general. In the following, we discuss these results under consideration of the current literature in the field of food consumption behavior and the effects of the COVID-19 pandemic. Additionally, we discuss limitations and directions for future research. Based on our findings, we further provide recommendations for practitioners in the field.

Overall, the findings show that relevant consumer attitudes, beliefs, and values that typically drive local food consumption [44] were largely unaffected by the first wave of the pandemic. As such, respondents showed stable scores before and after the COVID-19 outbreak for the majority of variables. These variables included, among others, *consumer ethnocentrism*, which depicts a consumer preference for local products, while avoiding imports in order to assist the local farmers and the economy. While national calls to support local suppliers during the crisis and consumers' experience of vulnerable global supply changes might have nourished expectations of attenuated ethnocentric tendencies among consumers, our data does not support such tendencies. Similarly, the perceptions of consumers regarding the the quality of domestic food, and their local identity remained stable. Regarding the role of ethical concerns in food choice, the literature has revealed mixed findings. Our study indicated no crisis-induced change in *green consumer values*, in line with Mertens and colleagues' study [50]. What changed was the *perceived authenticity* of the local agri-food sector. Consumers accordingly placed less relevance on how authentic the suppliers were after the first wave of the pandemic, which could be explained by an increase in the relevance of other characteristics, such as the resilience of suppliers in times of crises [6–8]. Correspondingly, the stability observed in consumer perceptions might be related to the stability of the supply in Austria. Accordingly, the food supply chains in the Global North showed a high resiliency and could, thus, ensure stable access to food shortly after the first shocks in response to the outbreak of the pandemic [13,14,67]. The grocery store shelves were, for example, replenished as consumers reduced the volume of their food purchases after initial stockpiling [67]. This suggests that consumers were not severely affected by downstream effects which move from suppliers, that experienced disruptions, to consumers [13,39].

By contrast, consumers' *price consciousness* for food, on average, decreased in our sample in response to the first wave of the pandemic. The first lockdown and imposed mobility restrictions triggered panic buying and hoarding behaviors in consumers [6,28,33], which might have overshadowed their consciousness of the price of the purchased products in the short run. In the long run, however, the literature expects an increase in price consciousness [6]. For the Austrian context, data from the Austrian Corona Panel Project (ACCPP) suggested that during the lockdown in April 2020 household income losses averaged about 12% [68], which might, at least for some consumer segments, increase price consciousness for food. This development is expected to be aggravated by increasing food prices and consumer price indices as a consequence of the war in Ukraine [2,69] and, thus, opens an avenue for future research.

With regard to the role of the domestic agri-food sector during the crisis, our findings depict a positive appraisal by consumers. Consumers perceived the sector to be a reliable partner during the pandemic. This finding is in line with the above argument on a lack of experienced downstream effects on consumers during the first wave of the pandemic. In the further course of the pandemic, the Austrian agri-food sector continued to ensure a stable food supply and maintained high self-sufficiency rates of about 90% for grain and potatoes and about 50% for fruits and vegetables [32]. Within the local agri-food sector, value-based modes of production and consumption, such as community-supported agriculture (CSA) and alternative and local food systems (ALFS), became more visible [60]. While these initiatives are still a niche in Austria [70], the positive consumer perceptions about the domestic agri-food sector, in general, might create an impetus for such modes of production and consumption. Our study finally revealed that consumers trust the sector to manage future crises.

At the time of writing, such a future crisis affecting the global food supply has sadly become reality. With the Ukraine war, global dependencies have become even more visible [71]. The current trade policies, export restrictions, and increasing energy prices are further generating downstream effects in the form of higher food prices [72,73]. The war in Ukraine, post-COVID-19 macroeconomic developments, such as global inflation pressures, as well as EU spring weather developments pose an additional major threat to global food security [69] by repeatedly exposing the vulnerability of global food supply chains [2,74,75]. Consumers might, therefore, be affected by downstream effects differently compared to when the COVID-19 pandemic first started in early 2020, e.g., in the form of limited availabilities of respective products. The implications of this potential momentum for consumer perceptions, the actual consumption of local food, and potentially newly emerging drivers present an interesting avenue for future research that considers ongoing changes in the field of food consumption at an individual level in response to macro-level disruptions. Specifically, subjective perceptions of the different actors were recently identified as a relevant aspect to be captured to better understand how agricultural systems can change over time [15].

From a practical perspective, the demonstrated resilience of the local suppliers and perceptions of them as reliable partners during the first wave of the pandemic could be emphasized for the promotion of local food. Such messages might also decrease potential consumer concerns about food supply in times of current or future crises. The fast pace at which food supply chains were able to adjust to the initial shocks and enforce their supply chain resilience against further disruptions can help to shape long-run consumer confidence in those supply chains [6]. Our results further echo results from the literature suggesting that a new relationship of trust between consumers and local sales networks could be established as an effect of the proven resilience of local suppliers during the pandemic [14]. A reliable and robust relationship is central to enhancing resilience [15]. A key consumer data 2020 assessment of the European Union (EU), for example, already revealed that in the EU on average 81% shopped closer to home and supported local businesses, whereas, in Austria, this was above average with 92% following the outbreak of the COVID-19 pandemic [76]. As the outbreak of the pandemic also led to an increase in direct sales to consumers via farm shops or delivery services [25], the sales channel could present a promising starting point for this type of promotion.

After the first outbreak of the pandemic, consumers experienced several subsequent waves of COVID-19, which imposed lockdowns and mobility restrictions upon them [77]. Such an early assessment provides insights into consumers' initial reactions to shocks bringing novel situations. These insights on how consumers receive downstream effects or send upstream effects resulting from initial disruptions might be valuable for policymakers in designing the first responses to those shocks. However, consumers might have continued to adapt consumption patterns which can further change their perceptions (see [62]). As our study was carried out at an early stage of the pandemic, it did not allow us to make predictions about the stability of consumer perceptions through the course of the pandemic and its changing conditions. Conducting replication studies in later stages, in which lockdowns and mobility restrictions are no longer in place in Austria, would allow the determination of the implications of the COVID-19 pandemic from a long-term perspective. We, therefore, recommend future research to replicate studies from early stages so that the findings about changes can be validated over time.

Aside from limitations stemming from the assessment period, there are limitations due to the geographical study context. Austria is a country of the Global North, which showed high resiliency amidst the outbreak of the pandemic and could, thus, ensure stable access to food shortly after initial disruptions [13,14,67]. It is also an example of a country with a high self-sufficiency rate for fruits and vegetables [32]. Thereby, the findings of our study are specific to the Austrian market and indicative of countries with similar characteristics. It might, however, not be representative of other countries, for example, in the Global South.

The upstream and downstream effects of disruptions that affect food supply chains in the case of more destabilized contexts could thus be explored within future research.

In sum, our findings contribute to a better understanding of crisis-induced consumer perceptions and attitudes relevant to local agri-food sectors. They showed that consumers appraised the stability of food supply which they attributed to the local agri-food sector. In the lack of experiencing negative downstream effects, such as supply shortages, their values and dispositions driving local food consumption remained stable at large, while price consciousness for food decreased. The effect of crises causing negative downstream effects, such as empty shelves, on these drivers and perceptions is a relevant question to be tackled by future research.

**Author Contributions:** Conceptualization, P.R. and L.M.W.; methodology, P.R. and L.M.W.; software, P.R. and L.M.W.; validation, P.R. and L.M.W..; formal analysis, P.R. and L.M.W.; investigation, L.M.W.; resources, P.R. and L.M.W.; data curation, P.R. and L.M.W.; writing—original draft preparation, L.M.W.; writing—review and editing, P.R. and L.M.W.; visualization, L.M.W.; supervision, P.R.; project administration, P.R.; funding acquisition, P.R. All authors have read and agreed to the published version of the manuscript.

**Funding:** This research was partially funded by "Österreichische Hagelversicherung".

**Institutional Review Board Statement:** Ethical review and approval were waived for this study, due to the full anonymity of interviewees and as no sensitive data were collected. Personal data (e.g., age, gender) cannot be traced back to individuals. The recruiting of and information provision to interviewees is in accordance with the ethical standards of the respective Panel Provider.

**Informed Consent Statement:** Informed consent was obtained from all subjects involved in the study.

**Data Availability Statement:** The data presented in this study are available on request from the corresponding author. The data are not publicly available to guarantee maximum data security and privacy of respondents.

**Acknowledgments:** We would like to thank the respondents who participated in our survey. We also want to acknowledge the support of our student assistants, Caroline Kunesch and Katharina Spöck, in the project presentation, which facilitated the data analysis.

**Conflicts of Interest:** The authors declare no conflict of interest.

## Appendix A

**Table A1.** Measurement items to assess constructs of consumer perceptions about local food consumption.

| Constructs | Measurement Items |
|---|---|
| Consumer ethnocentrism [45] | One should not buy imported food because it hurts Austrian farmers.<br>It is not right to purchase imported food because it causes the loss of jobs in Austria.<br>One should only buy local food.<br>I always prefer Austrian food over imported products. |
| Healthiness bias Adapted from [46,63] | Local foods are more nutritious.<br>Local foods are healthier.<br>Local foods are more environmentally friendly.<br>Local food tastes better.<br>Local foods have higher standards.<br>Local foods are more strictly controlled. |
| Local identity [64] | My heart belongs to my local community.<br>I respect my local traditions.<br>I see myself as a local citizen.<br>I care about knowing about local events. |
| Authenticity [58] | Local agriculture produces food that is original.<br>Local agriculture puts authentic food on your plate.<br>With local agriculture, I know what I get.<br>Austrian food gives me a feeling of home. |

**Table A1.** *Cont.*

| Constructs | Measurement Items |
|---|---|
| Green consumer value [54] | It is important to me that the products I use do not harm the environment. I consider the potential environmental impact of my actions when making many of my decisions. My purchase habits are not affected by my concern for our environment. I am concerned about wasting the resources of our planet. I am not willing to be inconvenienced in order to take actions that are more environmentally friendly. |
| Price consciousness [65] | I buy groceries mainly when they are on sale. Price is the most important factor for me when choosing food. |

**Table A2.** Measurement items to assess constructs of consumer perceptions about local agri-food systems.

| Constructs | Measurement Items |
|---|---|
| Supply security and system relevance | Farmers played an important (system-relevant) role during the Corona crisis. Austrian agriculture was a reliable partner during the Corona crisis. Austrian agriculture was able to ensure a stable supply of food. |
| Profile during the pandemic | Continuation of agricultural farms throughout Austria. A sufficient supply of basic foodstuffs (such as milk, meat, and sugar) from Austria is also to be ensured in the future. Status of Austrian agriculture as a whole. The existing dependence on foreign food imports (such as eggs, fruits, and vegetables). Mandatory labeling of the origin of ingredients in processed foods (such as ready meals). |
| Trust in the local agri-food sector | How high is your trust that, in the event of similar crises in the future, there will be sufficient food available and no shortages in the food supply? |
| Financial vulnerability (Control variable) | How much have you personally been affected financially by the Corona crisis? |

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
