# Peer review of "Short-Term Effects of the COVID-19 Outbreak on Consumer Perceptions of Local Food Consumption and the Local Agri-Food Sector in Austria"

_agronomy, doi:10.3390/agronomy12081940_

Round 1

Reviewer 1 Report

The manuscript presents interesting results on local food consumption during crisis periods by comparing two periods of time. What differentiates from other studies that compare pre and post-covid is the fact that data were collected first in 2019 and later in 2020, thus reflecting better the reality at the time of data collection.

Below are my comments/questions:

1)     The analyses indicate that there are 6 hypotheses and not 1. In lines 166-168 these should be stated separated to avoid confusion. It is also easier to follow the results later if they are numbered.

2)     Results section: more needs to be revealed about how the constructs in Table 2 were created based on the items presented in the Annex. Same comment for constructs in Table 3. In appendix there are more constructs that in table 2 and 3. I assume constructs in Table 2 were already presented in another article (if so, please refer to it also in the results section, perhaps as a note to table 2). What about construct in table 3?

3)     What are the limitations of the study?

Reviewer 2 Report

The study analyzes the effects of the COVID-19 outbreak on consumer perceptions of local food consumption and the local agri-food sector in Austria, and establihs its conclusions based on the analysis of variables measured at two points in time: before the start of covid-19 pandemic (November 2019) and after the first wave of the pandemic (May-2019).

In my opinion, conducting the second survey in May 2020 offers irrelevant results in relation to consumer perceptions given that the pandemic lasted many more months and, therefore, these perceptions could have suffered significant variations. What does contribute knowing the change in perception at the end of the first wave of the pandemic , if there were four or five more waves? In my opinion, nothing. Why? because what is really relevant is to know whether once the sanitary measures are relaxed and in a situation of almost normality such as the one since April-May 2022, these perceptions about local food consumption have changed compared to the previous time of the pandemic. That is, if all these months of pandemic and confinement have modified purchase behavior, especially in relation to the consumption of local products.

I encourage the authors to repeat the survey considering the current situation of the pandemic (July-2022) and compare them with the data obtained in November 2019. With this temporary difference and considering that the pandemic is controled, the results obtained will allow us to affirm whether there has really been a change in the perception of Austrian consumers towards local food consumption and the local agri-food sector in Austria

Reviewer 3 Report

The manuscript entitled “Effects of the COVID-19 outbreak on consumer perceptions of local food consumption and the local agri-food sector in Austria” (agronomy-1850283) was aimed at identifying local food consumption changes in terms of the Austrian consumer preferences. This research was also aimed at explaining the role of local agri-food systems during times of crisis.

I propose the following with the purpose of improving the quality of the manuscript:

(1) The concept of resilience is not sufficiently explained, especially in relation with the local agri-food sector. It is only briefly mentioned in the abstract, introduction and conclusions section. Thus, I suggest expanding the manuscript with: (i) a description of resilience, (ii) how resilience applies to the local agri-food sector, and (iii) an explanation regarding the reasons why the resilience analysis is important in the context of this research.

(2) In the first part of the paper, I suggest emphasizing better the gap identified in the literature, as well as highlighting the novelty factor of this manuscript. This is partially explained in the last paragraph of the manuscript, at the end of the paper, but it needs more attention in the beginning.

(3) The discussion section should be much more focused towards explaining local food consumption changes in terms of the Austrian consumer preferences rather than tapping into the impacts of war between Ukraine and Russia in terms of possible food consumption changes – they are also welcome, but the focus should be switched.

(4) The limitations of this empirical study should be explained in the Conclusions section.

(5) Please correct the reference style from line 280 in accordance with the MDPI's guidelines.

Round 2

Reviewer 1 Report

The manuscript was improved based on recommendations. Just a minor change needed in line 273: "First, we tested our hypotheses (H1a-f)" should be "First, we tested our hypotheses (H1-6)"

Author Response

We are happy to hear that you see the manuscript as improved. Thank you again for your recommendations and for making us aware of the mistake regarding the hypothesis numbering.
We now adjusted the numbering  accordingly in Line 270.

Reviewer 2 Report

Although it is established as a limitation that the study was carried out at an early stage of the pandemic (first wave), I think that it is necessary that in the title of the work as well as in the abstract and the introduction, that aspect is specified in some way.
